# The genomic landscape of molecular responses to natural drought stress in *Panicum hallii*

John T. Lovell[1,2], Jerry Jenkins [1], David B. Lowry[3], Sujan Mamidi [1], Avinash Sreedasyam [1], Xiaoyu Weng[2], Kerrie Barry[4], Jason Bonnette[2], Brandon Campitelli[2], Chris Daum[4], Sean P. Gordon [4,8], Billie A. Gould[3], Albina Khasanova[2], Anna Lipzen[4], Alice MacQueen [2], Juan Diego Palacio-Mejía [2], Christopher Plott[1], Eugene V. Shakirov[2,5], Shengqiang Shu [4], Yuko Yoshinaga[4], Matt Zane[4], Dave Kudrna[6], Jason D. Talag[6], Daniel Rokhsar[7], Jane Grimwood[1], Jeremy Schmutz [1,4] & Thomas E. Juenger[2]

Environmental stress is a major driver of ecological community dynamics and agricultural productivity. This is especially true for soil water availability, because drought is the greatest abiotic inhibitor of worldwide crop yields. Here, we test the genetic basis of drought responses in the genetic model for $C_4$ perennial grasses, *Panicum hallii*, through population genomics, field-scale gene-expression (eQTL) analysis, and comparison of two complete genomes. While gene expression networks are dominated by local cis-regulatory elements, we observe three genomic hotspots of unlinked trans-regulatory loci. These regulatory hubs are four times more drought responsive than the genome-wide average. Additionally, cis- and trans-regulatory networks are more likely to have opposing effects than expected under neutral evolution, supporting a strong influence of compensatory evolution and stabilizing selection. These results implicate trans-regulatory evolution as a driver of drought responses and demonstrate the potential for crop improvement in drought-prone regions through modification of gene regulatory networks.

[1] Genome Sequencing Center, HudsonAlpha Institute for Biotechnology, Huntsville 35806 AL, USA. [2] Department of Integrative Biology, The University of Texas at Austin, Austin 78712 TX, USA. [3] Department of Plant Biology, Michigan State University, East Lansing 48824 MI, USA. [4] Department of Energy, Joint Genome Institute, Walnut Creek 94598 CA, USA. [5] Institute of Fundamental Medicine and Biology, Kazan Federal University, Kazan 420008, Russia. [6] Department of Ecology and Evolutionary Biology, University of Arizona, Tucson 85719 AZ, USA. [7] Department of Molecular and Cell Biology, University of California, Berkeley 94720 CA, USA. [8] Present address: Zymergen Inc, Emeryville 94608 CA, USA. Correspondence and requests for materials should be addressed to J.T.L. (email: jlovell@hudsonalpha.org) or to J.S. (email: jschmutz@hudsonalpha.org) or to T.E.J. (email: tjuenger@austin.utexas.edu)

Drought is the greatest abiotic determinant of agricultural yield[1] and a key regulator of both ecological net productivity and carbon sequestration[2,3]. Unlike annual plant species, which can escape environmental stress through flowering time evolution, perennial plants must persist through periods of drought. While constitutive drought tolerance can lead to slower growth and lower yields, there is considerable genetic variation in the physiological nature and efficacy of facultative responses to soil moisture variation among plants[4,5]. Leveraging such genotype-by-environment interactions (GxE), especially in response to drought, is key to crop improvement.

Despite the central role that GxE plays in adaptation and plant productivity, the genetic basis of evolved differences in stress responses are poorly understood, except in a handful of laboratory model systems. However, it is clear that gene expression networks play particularly important roles in the evolution of physiological GxE as transcription factors and other regulatory elements are often environmentally induced[6].

Regulatory elements fall into two main categories: distant trans-acting modifiers (e.g. transcription factors) and local cis-regulatory elements (e.g. promoter or coding sequence variants). Selection works most efficiently on traits that act in isolation; therefore, cis-elements, which typically regulate a single gene, may be particularly important in adaptive evolution[7]. Conversely, trans factors may cause correlated expression variation among many downstream genes, which should increase interference among loci and reduce the adaptive potential of global trans-regulatory evolution[8,9]. To this end, many global regulatory elements evolve much more slowly than their target sequences[10–12]. Combined, cis- and trans-regulatory elements shape the gene expression landscape and form the basis of environmental stress responses. Therefore, defining the regulatory elements that lead to evolved differences in stress responses will improve our understanding of stress adaptation and the genetic basis of GxE.

To understand the genetic basis of drought stress tolerance in the context of perenniality, we have developed *Panicum hallii* as a genomic model for Panicoid grasses[13]. The Panicoideae is a diverse subfamily of predominantly perennial warm-season grasses with efficient C₄ photosynthesis[14], which imparts drought-tolerance and some of the highest biomass production among plants[15]. Not surprisingly, the Panicoideae encompasses the most promising bioenergy feedstock crops, including switchgrass, *Sorghum*, big bluestem, *Miscanthus*, and sugar cane. The geographic distribution of *P. hallii* spans a massive moisture availability gradient: the lowland variety (*P. hallii* var. *filipes*; hereon *filipes*) inhabits riparian and coastal sites with >100 cm of annual precipitation while upland (*P. hallii* var. *hallii*; hereon *hallii*) populations are found across southwestern North America[16] in both desert (<20 cm annual precipitation) and semi-arid habitats. The extensive physiological diversity[13] of *P. hallii* and its close evolutionary relationship to key crops make it an ideal genetic model for biotechnology development in perennial biofuel feedstocks.

Here we present a set of experiments and analyses that dissect the genetic basis of *P. hallii* drought responses, connecting DNA sequence variation to leaf-level physiology. We first assess the demographic history and population structure of the upland and lowland varieties of *P. hallii*. To understand the scale of genomic divergence between varieties, we compare complete de novo genome assemblies and annotations for a single genotype of each variety. We then explore the genetic basis of evolved drought-responsive gene expression in an F₂ mapping population. Through eQTL mapping and comparative genomics, we demonstrate that trans-regulatory elements and transcription factor binding site evolution are key contributors to molecular drought responses in *P. hallii*. Combined, these results and

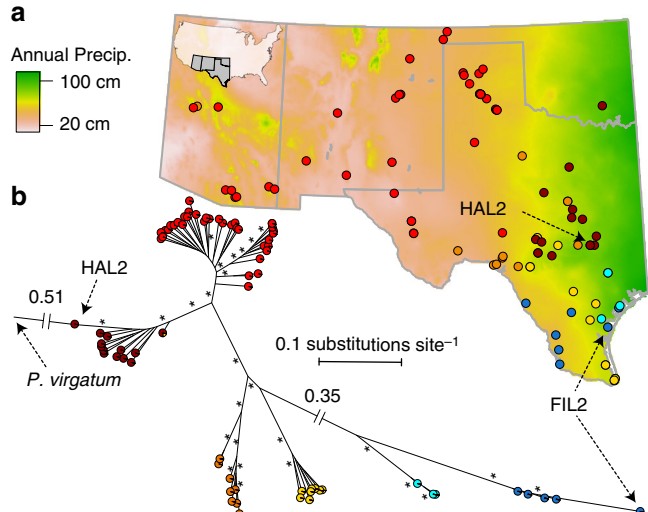

**Fig. 1** Genetic structure and geographic distribution of *P. hallii*. One individual was deeply re-sequenced from 94 locations across southwest USA (**a**). Filled points represent the geographic location of the population where each individual was collected. Point colors represent genetic subpopulation assignments from STRUCTURE. The map background is cropped from the annual precipitation (BIO12) 2.5-minute raster (http://worldclim.org/version2) via a 10-minute United States of American state-boundary vector shapefile (http://naturalearthdata.com). In addition to between-variety divergence (*filipes*: blue, *hallii*: red/orange/yellow), there was strong population structure within varieties (**b**); proportional subpopulation assignment is shown in pie charts at the branch tips. Branch lengths are proportional to the number of substitutions per variable site, except the root branch and the internal branch connecting varieties, which are labeled with the respective lengths; branches marked * have bootstrap support >90%. Source data are provided as a Source Data file

genetic resources elevate *P. hallii* among the elite plant genomes, provide the genetic model for perennial feedstocks, and begin to decipher the complex drought-responsive genetic networks that have diverged between upland and lowland ecotypes.

## Results

**Demographic history of *P. hallii*.** Habitat divergence between *filipes* and *hallii* is likely due to both adaptation and historical demographic processes associated with paleo-climatic change in North America. To explore the recent evolutionary history of these varieties, we deeply re-sequenced (35x median sequence coverage) 15 *filipes* and 78 *hallii* natural accessions (Supplementary Data 1) spanning the USA geographic range of each variety (Fig. 1a). Demographic models (Supplementary Figure 1) indicated that *hallii* and *filipes* began evolutionary divergence >1 M years before present (ybp). This conclusion was confirmed by a molecular-clock based estimate of divergence of 1.08 M ybp. Furthermore, pairwise cross-coalescence, a measure of the degree of genetic divergence between populations[17], between any two *hallii* and *filipes* subpopulations (Fig. 1b) all declined below 0.25 at least 500k ypb (Supplementary Figure 1). These data indicate that the two *P. hallii* varieties have maintained substantial reproductive isolation over a period of intense climatic fluctuations, including at least the last two glacial-interglacial cycles.

Effective population size ($N_e$) of the largest two *hallii* subpopulations (inferred via genetic clustering, Fig. 1b) expanded following reduction of cross coalescence until 20–100k ybp, while $N_e$ of the largest *filipes* subpopulation remained relatively stable and ~50% smaller than any *hallii* subpopulations (Supplementary

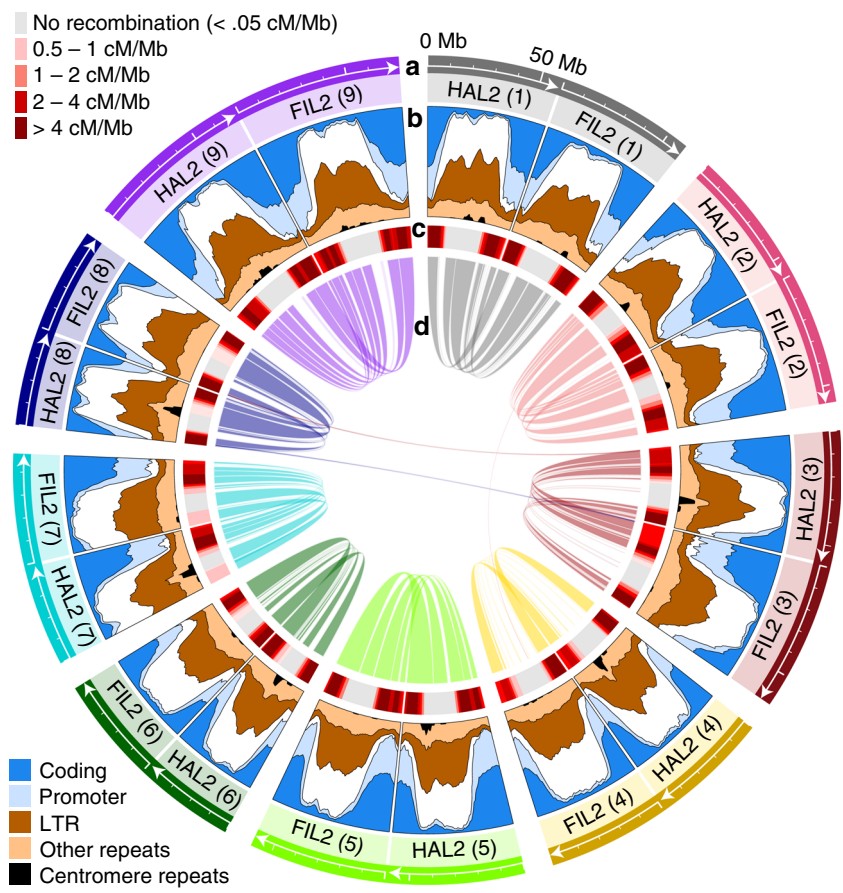

**Fig. 2** Synteny and structure of the *P. hallii* genome. The *P. hallii* genomes are characterized by highly genic chromosome arms and non-recombinant pericentromeric regions. The physical size and orientation of each chromosome (minor ticks at 10 Mb intervals) are plotted along the outer-most track (**a**). Gene models and repetitive sequences were annotated for both assemblies - the proportional representation (in 5 Mb overlapping windows) of five annotation categories, and unannotated intergenic sequence (white), are plotted in the second track (**b**). A heatmap of recombination rate (cM/Mb) is shown in the third track (**c**), where the darkest red represents >4% chance of a crossover event per $F_2$ individual per Mb. The interior links (**d**) connect 72 syntenic blocks, which cover 369.2 Mb (75.8%) of the total *HAL2* assembly and 254 Mb (84.5%) of the chromosome arms. Source data are provided as a Source Data file

Figure 2; Supplementary Data 2) over this period. However, $N_e$ of all subpopulations have contracted over the last ~50k years. Combined, these demographic patterns suggest that the current range of *hallii* is similar to, or smaller than its historical distribution and is not purely the product of a recent population expansion. Instead, between-variety habitat differentiation is consistent with historical range expansion of *hallii* into arid regions of the North American southwest.

**Development and comparison of two complete de novo genomes**. To test for a genetic signal of drought-associated evolution, we developed complete de novo reference genomes for the basal *P. hallii* accession (*var. hallii HAL2*) and a representative *var. filipes* accession (*FIL2*) (Figs. 1b, 2a–d). These Pacific Biosciences single molecule-based chromosomal assemblies contain 484.6 Mb (99.4% of assembled sequence in chromosomes, contig N/L50 = 15/8.3 Mb) and 508.0 Mb (94.8%, 117/1.1 Mb) of total *HAL2* and *FIL2* sequence, making them among the most complete plant genomes ever assembled (Supplementary Notes 1–2; Supplementary Figure 3; Supplementary Tables 1–4). Critically, these genomes exhibit near-perfect chromosome-scale synteny (Fig. 2d, Supplemental Note 3; Supplementary Figure 4, Supplementary Table 5; Supplementary Data 3). A total of 475 kb (<0.1%) of the *HAL2* genomic assembly was not collinear with *FIL2*, including a

0.18 Mb translocation on Chr09 and a 0.295 Mb translocation between *HAL2* Chr02 and *FIL2* Chr04 (Supplementary Table 5). Additionally, a duplication was present on the proximate telomere and adjacent 2.22 Mb region of Chr03 and Chr08 in both genomes. This duplication is also present in *Sorghum bicolor* (Supplementary Figure 4), indicating that it is ancient and ancestral to *Panicum*.

It is possible to leverage such extensive synteny to infer sequence evolution and presence-absence variation (PAV) among orthologous clusters of genes. This analysis was performed via our GENESPACE pipeline (Supplementary Note 3), which employs a multi-species orthologous gene network construction approach constrained within collinear sequence blocks. GENESPACE allows for construction of outgroup-rooted gene networks and sequence alignments within duplicated (e.g. proximate Chr03 and Chr08) and single-copy regions. Critically, by informing orthology networks with outgroup sequences, we can determine whether gene annotations without orthologous sequences between *HAL2* and *FIL2* are derived or lost in each genome. Overall, the majority of high-confidence gene models (41,017 genes; 80.1%) existed as either single copy ('1:1') orthogroups or orthogroups with two or more members in at least one genome ('1:2 +', '2 +:1' or '2 +:2 +'; Table 1; Supplementary Data 4). However, 10,176 genes (19.9% of all gene models) lacked

**Table 1 Summary of CDS orthology between *HAL2* and *FIL2***

| Category (*HAL2*: *FIL2*) | *n*. genes | % Low support transcripts | % Expressed F₂ leaf |
|---|---|---|---|
| 1:1 Orthologs[a] | 34,396 | 7.8 | 74.6 |
| 1: 2+ Orthogroup[b] | 548 | 36.7 | 42.3 |
| 2+: 1 Orthogroup | 484 | 39.0 | 38.6 |
| 2+: 2+ Orthogroup | 5589 | 23.2 | 51.8 |
| Private[c] to *HAL2* | 4455 | 77.7 | 20.6 |
| Private to *FIL2* | 4971 | 73.0 | 19.7 |
| Present[d]: un-annotated[e] | 160 | 84.3 | 7.3 |
| Un-annotated: present | 136 | 88.6 | 4.6 |
| Present: pseudogene[f] | 209 | 38.0 | 36.5 |
| Pseudogene: present | 212 | 38.4 | 43.7 |
| Present: deleted[g] | 19 | 52.6 | 35.7 |
| Deleted: present | 14 | 57.1 | 38.9 |

Orthology, un-annotated, and presence/absence are inferred via the GENESPACE pipeline (Supplementary Note 3). The gene-annotation category (where *HAL2* category precedes ':' and *FIL2* follows) and number of genes found therein are presented in the first two columns. Annotation confidence score (Supplementary Note 2) is calculated via the degree of homology, gene expression, and PFAM support. Gene models that did not satisfy these criteria were considered low support. To test for expression in leaf tissue in our field experiment, we counted transcript abundance in an *HAL2-FIL2* F₂ population. Expressed genes had at least one count in ≥10% of the F₂ population and mean counts >5, after excluding libraries with 0 counts.
[a]Orthologs are pairs of HAL2 and FIL2 gene models, where a single gene model from each genome is represented in an orthofinder orthogroup.
[b]Other orthogroups contain two or more gene models from one or both *P. hallii* genomes.
[c]Private genes are found in single-gene orthogroups without representation of any outgroup (*S. viridis* and *S. bicolor*) sequences.
[d]Present genes are found in orthogroups that contain one or more outgroup genes, while [e]un-annotated genes have sequence with >90% coverage in the alternative genome assembly but no gene annotation.
[f]Pseudogenes have >10% and ≤90%, and [g]deleted sequences have ≤10% similar sequence coverage in the alternate genome assembly.

annotated orthologous sequence in either *HAL2* or *FIL2* (Table 1; Supplementary Data 5). These PAV genes may be the product of annotation support thresholding, pseudogenization or true sequence deletion in one genome.

To evaluate the evolution of these PAV genes, we compared DNA variation between annotated coding DNA sequence (CDS) of present genes with the unannotated assembly sequence of syntenic orthologous regions in the other genome. The majority of PAV genes were private to either *HAL2* or *FIL2* and never had orthologous sequences in any outgroups (Table 1, Supplementary Data 5). Other studies find that such private genes are generally of low quality[18] and may not represent novel derived CDS, but instead are simply due to weak gene evidence that only survives thresholding in a single genome[19]. Indeed, private genes were 9.6x more likely to be low confidence, with low homology or transcript support, than 1:1 orthogroup genes.

To understand sequence variation underlying the remaining PAV genes, which also had at least one outgroup sequence in the network, we categorized the degree of sequence variation between CDS and un-annotated DNA sequence of the alternative genome into un-annotated (>90% sequence identity), pseudogenized (90–10% identity), and deleted (< 10% identity) groups (Table 1). Like the private gene models, the 296 un-annotated gene models were much more likely to be low confidence than genes with 1:1 orthologs (Fisher's test odds ratio = 11.0 ×, $P < 1 \times 10^{-16}$, Table 1, Supplementary Data 6). Combined the un-annotated and private PAV genes are very likely to be the product of annotation support thresholding and not true sequence deletion or pseudogenization. The remaining 454 pseudogenized and deleted gene models were 2.49x more likely to be high confidence than the private and un-annotated genes (Fisher's test $P < 1 \times 10^{-16}$), indicating that pseudogenized and deleted genes may represent biologically relevant presence-absence variation.

To confirm the quantitative genetic effects of the observed PAV, we counted expression of each unique allele across the two genomes in leaf tissue of an F₂ population (Table 1). These results largely mirror inference based on gene model confidence: private and un-annotated genes had weak expression, while pseudogenized and deleted gene models showed similar levels of expression as those within orthologous gene networks. Combined, the relative rarity of such PAV indicates that small insertion/deletions

(INDELs) and single-nucleotide polymorphisms (SNPs), and not large sequence deletions, represent the vast majority of molecular evolution between *HAL2* and *FIL2*. Furthermore, protein coding DNA sequence (CDS) gain and loss is not common despite the >1 My divergence time between *HAL2* and *FIL2*. Such conserved gene content and synteny despite significant sequence divergence (mean $\pi = 0.0195$) makes *P. hallii* an ideal system to test hypotheses about sequence evolution, selection, and molecular adaptation.

**Genetic mapping to dissect regulatory network evolution**. To map the genetic basis of physiological divergence between varieties, we conducted a large-scale field drought experiment where 25 *HAL2* and 34 *FIL2* replicates and 243 *FIL2xHAL2* F₂ genotypes were subjected to a month-long natural drought. Half of the plants were watered 24 h prior to harvest (recovery treatment), while the remainder were harvested under existing drought conditions. At the physiological scale, midday leaf water potential (a measure of plant water status) was more responsive to re-watering among *HAL2* than *FIL2* plants ($\chi^2_{df = 1} = 4.01$, $P = 0.045$, Supplementary Figure 5). This suggests that the arid-adapted *hallii* was more sensitive to temporally variable water resources, a trait that is likely advantageous in desert ecosystems marked by drought but punctuated by brief periods of high soil moisture[20].

Physiological responses like leaf water potential are driven in part by the evolution of gene expression regulatory sequences[21,22]. To dissect the genetic basis of regulatory network divergence between *hallii* and *filipes*, we assayed total RNA abundance of the F₂ population (Supplementary Data 7), constructed a genetic map (Supplementary Figure 6), and subsequently conducted gene expression quantitative trait locus (eQTL) mapping. To determine significance of constitutive (QTL) and treatment-responsive (QTL*E) QTL for the expression phenotype of each gene, we compared NULL (expression ~E; no QTL, only drought treatment), additive (expression ~QTL + E), and full (expression ~QTL + E + QTL*E) QTL models via likelihood ratio tests.

While PAV and multi-copy orthogroups are important aspects of evolution between *HAL2* and *FIL2*, regulatory network inference of these genes is obscured by different copy numbers

**Table 2 Summary of cis- and trans-eQTL effects**

| eQTL category | n. QTL | Mean LOD | Mean PVE (%) | CDS odds vs. no QTL | TFBA odds vs. no QTL | CDS odds vs. cis | TFBA odds vs. cis | CDS odds vs. trans | TFBA odds vs. trans |
|---|---|---|---|---|---|---|---|---|---|
| Cis (additive) | 7040 | 29.7 | 29.2 | 1.12[*] | 1.51[***] | --- | --- | 1.32[*] | 1.64[***] |
| Trans (add.) | 576 | 7.3 | 7.4 | 0.85[+] | 0.93 | 0.76[*] | 0.61[***] | --- | --- |
| Cis (GxE) | 2048 | 37.3 | 33.7 | 1.20[*] | 1.79[***] | 1.06 | 1.19[*] | 1.41[*] | 1.94[***] |
| Trans (GxE) | 738 | 6.6 | 7.3 | 0.95 | 0.75[*] | 0.85 | 0.49[***] | 1.12 | 0.81 |

Total number, mean LOD, and mean percent of phenotypic variance explained (PVE) are presented for each unique combination of additive/GxE and cis/trans eQTL. Each gene was also qualified as having significantly diverged coding DNA sequence (CDS) or promoter sequence evolution within transcription factor binding sites (TFBA). The Fisher's test odds of enrichment of evolution at either of these sites are presented against three backgrounds: genes without QTL (no QTL), genes with only cis-eQTL or only trans-eQTL. Therefore, odds >1 indicates cases where the test group has evolved more sequence variants between *HAL2* and *FIL2* than the background. Genes with both cis- and trans-eQTL are excluded for the enrichment tests. Significance codes: [***]P-value<0.00001, [*]P-value<0.01, [+]P-value<0.1.

of the focal gene. For example, some F$_2$ genotypes will simply lack a PAV gene model, which would bias inference of the strength and presence of eQTL. Therefore, we limited our eQTL analysis to only those high-confidence genes (which have both homology and transcript evidence) with one-to-one reciprocal best hit (RBH) orthologs between the two genomes. This test allowed us to count the total expression of 21,227 putative RBH orthologs (Supplementary Data 3) by summing allele-specific and shared transcript counts; 80.4% (17,061) of these had a mean of >1 raw count across the F$_2$ population. The voom[23]-normalized transcript counts from these genes were used as the independent phenotypic variables in our eQTL analysis. Combined, we found significant eQTL among 58% of the 17,061 genes with RBH orthologs between *HAL2* and *FIL2*.

The genetic architecture of regulatory variants can have profound consequences on the evolutionary processes that produce heritable genetic diversity[7,24]. Cis-elements in coding or promoter regions typically regulate the expression of a single, physically linked gene. In contrast, trans-elements, like transcription factors, can have global regulatory effects, which may constrain adaptation by increasing the likelihood of interference and limiting the efficacy of selection[8]. Due to the potential antagonistic effects of trans-regulatory elements, we expected drought-adaptive regulatory networks in *P. hallii* to be dominated by cis variants. Indeed, the 9,088 significant cis-eQTL on average explained 30.2% of total gene expression variation while the 1,314 trans eQTL explained just 7.4% (Table 2).

To develop candidate sequence variants for cis-regulatory QTL, we conducted coding (Supplementary Data 9) and regulatory sequence (Supplementary Data 10) alignments of RBH orthologs between the *HAL2* and *FIL2* genomes. Cis-regulated genes were significantly more likely to have accumulated non-synonymous and other coding variants than genes without QTL (Table 2). This enrichment of potentially functional variants was much stronger within promoter regions, where genes with cis-eQTL were 1.5x more likely to have evolved differences in transcription factor binding affinity (TFBA) than genes without eQTL (Table 2). While it is likely that genes with differential expression are generally subject to weaker evolutionary constraint than those without evolved expression variation, these results show that cis-eQTL are generally driven by proximate regulatory loci, and not by genetically linked, but physically distant, regulatory loci.

Genes with only trans-eQTL are, by definition, not differentially regulated by local sequence variants. Consistent with such non-local effects, genes regulated by only trans-eQTL had slightly more conserved CDS (Fisher's test odds = 1.2x more conserved, $P = 0.08$) and similarly conserved TFBA (Fisher's test odds = 1.08×, $P > 0.1$) regions as genes without eQTL. Furthermore, since trans-regulated loci are targets of transcriptional regulatory elements, we expected transcription factor binding sites to be very conserved among genes with only trans-eQTL. Indeed, such genes

had 1.45x greater transcription factor binding site sequence conservation (Fisher's test $P < 1 \times 10^{-6}$) than genes with cis-eQTL, but less significantly conserved CDS regions (Fisher's test odds = 1.32×, $P = 0.003$, Table 2). These patterns were even more significant among genes with GxE QTL (Table 2). These results suggest that non-synonymous variants are less responsible for the evolution of expression regulation than promoter sequences and demonstrate sequence conservation of trans-acting transcription factors binding sites.

**Exploring the causes of trans-eQTL hotspots.** Differential adaptation across habitats is an example of a genotype-by-environment interaction (GxE) for fitness[4]. Therefore, loci that contribute to adaptation may be disproportionately environmentally responsive and exhibit trade-offs[25]. Despite their global rarity, trans-eQTL were more than four times as likely as cis-eQTL to have significant GxE effects (Fisher's test odds = 4.40, $P < 1 \times 10^{-16}$, Table 2), indicating a potentially adaptive role of trans-regulatory evolution between *P. hallii* varieties. Furthermore, there were three genomic hotspots where trans-eQTL were more common than cis-eQTL, which represented 3.1% of the physical genome sequence but 40.4% of all trans-eQTL (Fig. 3a–b, Table 3). Combined, these hotspots were responsible for 3.84× (Fisher's test $P < 1 \times 10^{-16}$) more GxE effects than all trans-eQTL outside this interval.

It is important to note that the eQTL hotspots we identified may not be caused by regulatory element evolution, but instead could be driven by the presence of a large physiological QTL that altered plant water status and caused downstream gene expression variation of many water status-responsive genes. To test this hypothesis, we collected leaf water potential (LWP), which is the best available field-scale proxy for plant water status[26], from the entire F$_2$ population at both pre-dawn and midday at the same time as RNA sampling. There were no significant QTL (lowest empirically-derived $P$-value = 0.27, Supplementary Table 6) on any chromosome for either sampling period, indicating that genetic variation in plant water status does not in turn drive the physical clustering of trans-eQTL. Instead, the trans-eQTL hotspots and their significantly elevated environmental sensitivity suggest that pleiotropic mutations in transcription factors may be a major source of regulatory variation.

We searched for candidate genes by investigating the distribution of allelic effects among QTL mapping to each hotspot[27]. For example, expression of all but two genes regulated by trans-eQTL in the *3a primary hotspot were driven by plasticity of the *FIL2*, but not the *HAL2* allele (Fig. 3c). Within this 964 kb interval, three genes had significant cis-eQTL (Supplementary Data 8) and elevated non-synonymous substitution rates (Supplementary Data 9). The most promising candidate, *ABO3* (*P. hallii* ortholog of *A. thaliana* AT1G66600 - *ABA Overly*

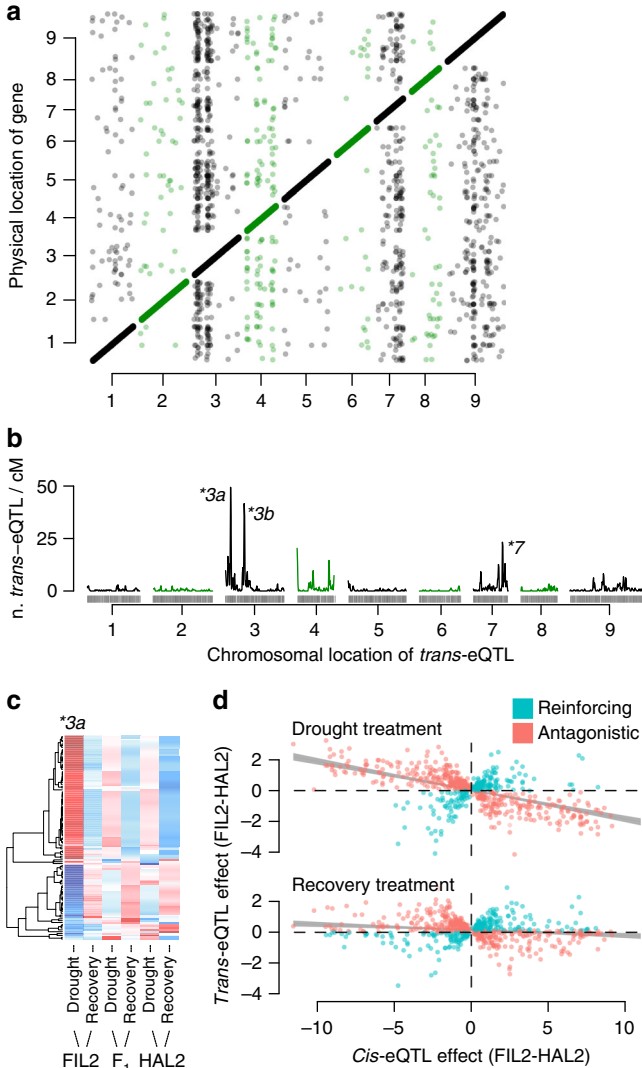

**Fig. 3** Distribution and effect of trans-eQTL. The position of each cis (on diagonal) and trans-eQTL (off diagonal) and the location of the proximate marker to the physical gene position are plotted (**a**). Points and line segments are colored green/black to distinguish adjacent chromosomes. The density of trans-eQTL/cM along sliding windows were scored across the genome (**b**). The three strongest peaks (hotspots) are labeled (*). The allelic effects of the *3a trans-eQTL are plotted as heatmaps, where red colors indicate higher gene-scaled expression (**c**). The color scale of the heatmap corresponds to the scaled allelic effects of the *3a hotspot, where increasing blue and red color intensity indicate stronger negative and positive allelic effects respectively. White cells have a scaled allelic effect of zero. We tested for biases of compensatory or reinforcing evolution between cis and trans eQTL. Overall, there was significant biases towards antagonistic effects, however this bias was much stronger in the drought than recovery treatment (**d**). Source data for panels **a** and **d** are found in Supplementary Table 8. Source data for panels **b**, **c** are provided as a Source Data file

sensitivity of non-functional *ABO3* alleles in *A. thaliana*. It is important to note that eQTL analysis cannot definitively identify the causal loci driving trans-eQTL hotspots but instead provides candidates for future validation. However, the observed trans-eQTL colocalization clearly demonstrates the central role of global regulatory elements in evolved molecular drought responses.

**Signatures of selection among cis–trans regulatory networks.** Elevated GxE among trans-eQTL, particularly within the hotspots, indicates possible selection on gene expression plasticity. To test the hypothesis of non-neutral regulatory network evolution, we examined the QTL effect distribution among the 684 genes with both cis- and trans-eQTL (Table 3). Under a neutral model, cis- and trans-regulatory elements should evolve independently, which produces a random distribution of trans- relative to cis-effect directionality (up/down regulated)[29]. However, directional and stabilizing selection produce correlated shifts among regulatory loci[29,30]. For example, if directional selection favors up-regulation of a gene, adaptive evolution of both cis- and trans-regulatory loci would increase expression; such positive correlations among loci are known as reinforcing QTL effects. Alternatively, if stabilizing selection favors maintenance of ancestral expression levels, non-adaptive evolution of increased expression may be compensated for by selection for regulatory repression at another locus. Such compensatory evolution promotes antagonistic (negatively correlated) cis–trans regulatory networks.

We observed a significant genome-wide bias towards antagonistic effects (Fig. 3d, Fisher's test odds = 1.3, P = 0.015) and a very strong signal in the drought treatment, where allelic effects were greater than two times more likely to be antagonistic than under neutral expectations (Fig. 3d, Fisher's test odds = 2.15, P < $1 \times 10^{-12}$). However, the relative strength of reinforcing and antagonistic effects was highly dependent on the position of the trans-eQTL (Table 3). For example, the *3a hotspot was enriched in reinforcing effects relative to all other trans-eQTL; however, the proportion of reinforcing effects at *3a was not significantly different than the neutral expectation of 1:1 (Table 3). In contrast, the *3b hotspot was disproportionately represented by genes with antagonistic cis-trans effects in both treatments relative to the background and neutral expectation (Table 3). These results indicate a genome-wide prevalence of antagonistic effects, which is typical of stabilizing selection on expression regulation.

**Validating the physiological effects of a trans-eQTL hotspot.** Given the large number of genes regulated and the elevated rate of GxE, the trans-eQTL hotspots clearly drive variation in molecular responses within our drought-recovery experiment. In switchgrass, regulatory responses depended on the intensity and duration of drought[31], and our drought experiment is only a subsample of possible drought conditions. Therefore, we sought to validate the effects of the trans-eQTL hotspots in a broader context by exploring drought responses in a recombinant inbred line (RIL) population, derived from the $F_2$ used in the eQTL study (Supplementary Table 8). We assayed leaf relative water content (RWC) and chlorophyll content (SPAD) and conducted allelic contrasts at the proximate markers to each trans-eQTL hotspot peak position. Since RILs represent a random sample of genetic backgrounds, this analysis permits inference of causality when testing the effects of allelic differences at a locus.

Allelic variation at the *3b hotspot never significantly affected any treatment-trait combination, and the *7 hotspot had only one significant association (RWC under drought, Supplementary Table 8). However, variation at the *3a hotspot marginally

*Sensitive 3*, *PhHAL.3G069100/Pahal.3G071300*), is a well-documented drought-responsive WRKY transcription factor. Knockout alleles of *ABO3* in *A. thaliana* induce extreme sensitivity to drought[28]. RT-qPCR revealed that the *FIL2 ABO3* allele is 27.7x downregulated relative to the *HAL2* allele across both well-watered and drought conditions (Supplementary Table 7). The elevated drought-response caused by the weak *FIL2* allele is consistent with the extreme environmental

**Table 3 Summary of direction of effects for genes with both cis- and trans-eQTL**

| Hotspot ID | n. | Coordinates (Mb) | Treatment: drought | | | | Treatment: recovery | | | |
|---|---|---|---|---|---|---|---|---|---|---|
| | | | n. reinforce | n. antag. | Odds v. background | Odds v. neutral | n. reinforce. | n. antag. | Odds v. background | Odds v. neutral |
| 3a | 206 | 0.1–7.3 | 43 | 42 | 2.27[**] | 1.02 | 43 | 42 | 1.44 | 1.02 |
| 3b | 183 | 12.3–14.8 | 18 | 50 | 0.8 | 0.36[*] | 20 | 48 | 0.59[+] | 0.42[*] |
| 7 | 127 | 34.3–39.7 | 32 | 26 | 2.73[**] | 1.23 | 29 | 29 | 1.4 | 1 |
| Overall | 1314 | NA | 217 | 467 | – | 0.46[**] | 296 | 388 | – | 0.76[*] |

The total number of QTL with reinforcing (n. reinforce) and antagonistic (n. antag.) effects are presented for each treatment and split by the position (if the trans-eQTL is in one of the three hotspots). Fisher's exact tests for imbalance between antagonistic and reinforcing effects were conducted within each treatment with two NULL hypotheses: (1) the total bias towards reinforcing effects among all genes with cis- and trans-eQTL (v. background) and (2) a 1:1 ratio, expected under purely neutral evolution (v. neutral). Odds presented indicate the ratio of the observed bias toward reinforcement relative to the NULL ratio at each trans-eQTL location. Significance codes: [**]$P$-value<0.001, [*]$P$-value<0.05, [+]$P$-value<0.1.

affected RWC plasticity ($t = -1.95$, $P = 0.052$), RWC in the drought treatment ($t = 1.55$, $P = 0.081$), and significantly affected SPAD in both recovery ($t = 2.42$, $P = 0.016$) and drought treatments ($t = 2.02$, $P = 0.044$). Individually, each of these effects are not highly significant. Yet, when taken together we observe strong support for functional effects of allelic variation at the *3a hotspot with a Fisher's combined test ($\chi^2_{df=12} = 27.3$, $P = 0.0068$, Supplementary Table 8). Combined, the allelic effects at the *3a hotspot are consistent with those of ABO3 (lower plasticity, higher constitutive RWC and SPAD in *HAL2*) and demonstrate the predictive potential and multi-environment effect of trans-acting transcriptional regulatory elements.

## Discussion

The intersection of physiology, quantitative genetics, and whole genome sequencing holds great promise for understanding the complex interaction between genetic variation and the environment[31]. Using de novo whole genome assembly, large-scale field experimentation, and physiological genomic techniques in *P. hallii*, we were able to precisely map global regulatory loci and infer candidate variants. These experimental resources allow us to better infer the genetic basis of complex traits in plant adaptation, predict responses to current and future climatic stress, and develop a strategy for drought-responsive biotechnology in bio-fuel breeding programs.

## Methods

**Genome sequencing assembly and annotation**. We sequenced the *Panicum hallii* var. *hallii* and var. *filipes* genotypes HAL2 and FIL2 using a whole genome shotgun strategy and standard sequencing protocols. Sequencing was conducted on both Illumina (HISeq) and Pacific Biosciences (SEQUEL) platforms at the Department of Energy Joint Genome Institute (JGI, Walnut Creek, CA, USA) and the HudsonAlpha Institute for Biotechnology (Huntsville AL, USA).

Sequencing effort for HAL2 (Supplementary Note 1) included one 800 bp insert $2 \times 250$ fragment library ($150 \times$) and a total of 89.5x of PACBIO reads (average subread length 11.8 kb). FIL2 sequencing was similar (Supplementary Note 1), with one 500 bp insert $2 \times 150$ fragment library ($100 \times$) and a total of 95.87x of Pacific Biosciences reads (average subread length 9.6 kb). Both the FIL2 and HAL2 assemblies were performed using MECAT[32] and polished using QUIVER[33].

We built a 325,613-marker map from shallow resequencing of the RIL population[34]. This marker order was used to identify mis-joins in the assembly. A total of 115 mis-joins were identified in the FIL2 assembly and only 1 mis-join in HAL2. Scaffolds were then oriented, ordered, and joined together. A total of 1246 (FIL2) and 119 (HAL2) joins were applied to the broken assemblies to form the final releases consisting of 9 chromosomes each. Short redundant sequence from contig ends were aligned to one another and collapsed when appropriate. A total of 18 (HAL2) and 450 (FIL2) repeated adjacent contig pairs were identified and collapsed. For the FIL2 release, a set of 30,315 (430.3 Mb) targeted clone sequences (Illumina), along with a set of 704,618 (3.74 Gb, 6.9x coverage) MOLECULO reads were used to patch 60 sequence gaps. Finally, Illumina reads were employed to correct any remaining MECAT consensus calling errors (e.g. homozygous SNPs and INDELs).

PERTRAN (Supplementary Note 2) and PASA[35] were used to produce 92,211 (HAL2) and 111,325 (FIL2) transcript assemblies from ~1100 M (HAL2) and ~1200 M (FIL2) $2 \times 150$ paired-end Illumina RNA-seq reads. Repetitive DNA

elements were identified de novo with repeatModeler[36]. For mapping and genome comparisons, we soft-masked the genomes using repeatMasker[37], with ancestral repeats from RepBase and the repeat annotations from repeatModeler output. Loci were determined by transcript assembly alignments and/or EXONERATE (https://github.com/nathanweeks/exonerate) alignments of proteins from *Arabidopsis thaliana*, soybean, Kitaake rice, sorghum, foxtail millet, *Brachypodium distachyon*, grape and Swiss-Prot proteomes. Gene models were predicted by homology-based methods, FGENESH + /_EST[38] GenomeScan[39], and AUGUSTUS via BRAKER1[40]. The best-scoring predictions for each locus were selected using EST and protein support and penalized if overlapped with repeats. PASA was employed to add UTRs, splicing corrections, and alternative transcripts. High-confidence transcripts were called for loci with BLASTP Cscores >0.5, EST coverage, and <20% of CDS overlapping with repeats (if >20% overlap with repeats, only loci with Cscores >0.9 and homology coverage >0.70 were retained). Gene models with >30% TE domains (Pfam) were also culled. Finally, gene models with a short single exon (<300 BP CDS), without protein domain or with weak expression evidence were removed.

**Comparative genomics**. We used comparative genomic approaches to accomplish the following goals: (1) identify orthologous pairs of genes, (2) define the scale and causes of presence absence variation among gene annotations, and (3) understand the scale of synteny between the HAL2 and FIL2 genomes. Given these goals and the highly-repetitive and less conserved intergenic regions in plant genomes, we used a gene-level approach to whole-genome alignments. We ignored regions that were not in proximity to annotated gene models.

The GENESPACE pipeline (Supplementary Note 3) is applied to a set of de novo genomic assemblies and annotations. In short, GENESPACE conducts standard inference of orthology using the orthofinder[41] program but limits the search within known colinear (syntenic) blocks, generated by the multiple-collinearity inference program MCScanX[42]. This allows for the inference of orthology in duplicated chromosomal regions, as these appear as multiple distinct blocks in the alignments. In addition to pairwise peptide–peptide searches for orthologous gene groups, GENESPACE also conducts alignments against un-annotated genomic sequences (via BLAT[43] and EXONERATE) to discover the sequence identity of pseudogenized or otherwise un-annotated loci. The pipeline outputs alignments and some general sequence-divergence statistics for all orthogroup sequences among all genomes considered.

**F2 RNA sequencing and analysis**. HAL2, FIL2, and the $F_2$ mapping population were exposed to a short term recovery drought experiment following Lovell et al.[44]. In short, all plants experienced a natural 30-day drought at the Ladybird Johnson Wildflower Center (Austin, TX; 30.19° N, 97.87° W). Drought treatment plants were harvested on 5 July 2013, while recovery treatment plants received 4 L of water on 7 July 2013 and were harvested on 8 July 2013. For each plant, we measured midday leaf water potential (LWP, $\Psi_{leaf}$) with a Scholander-type pressure bomb (PMS Instruments, model 1000) between 11:00 and 13:00. All plants reached anthesis by 5 July 2013.

Leaf tissue harvest was conducted on 5 and 8 July 2013 between 11:00 and 13:00, where the most recent fully emerged leaf was immediately flash frozen with liquid nitrogen. For total gene expression assays, RNA (3 µg, RIN ≥ 5) was extracted from 50–200 mg of homogenized (Geno/Grinder, Spex SamplePrep) and DNase 1-treated leaf tissue with RNeasy Plant Mini kits (Qiagen). Total RNA libraries were prepared on a PerkinElmer SciClone NGS robotic liquid handling system using Illumina's TruSeq Stranded mRNA HT Sample Prep kit with 1 µg RNA per sample, and 10 cycles of library amplification PCR. Library quantification by KAPA Biosystem's next-generation sequencing library qPCR was accomplished on a Roche LightCycler 480 real-time PCR instrument. Sequencing was performed on the Illumina HiSeq 2000 sequencer and a TruSeq SBS sequencing kit (200 cycles, v3, following a $2 \times 150$ indexed run recipe).

To quantify expression, we mapped reads to concatenated genome assemblies and annotations of the HAL2 and FIL2 genomes. Uniquely mapping reads were

counted with STAR[45] – by definition these reads represent allele-specific expression (ASE). Shared (not allele-specific) reads were those that mapped to both orthologs; these were counted with the subread[46] featureCount function. Total counts, calculated as the sum of ASE and shared counts, were used for eQTL analyses. High-confidence gene models with reciprocal best hit one-to-one orthologs and average total counts ≥1 were retained for further analysis. Total counts were voom[23] normalized for eQTL analysis.

**eQTL analysis**. Expression quantitative locus mapping was performed in R/qtl[47]. It is common to perform eQTL via one-way single QTL scans for each gene expression trait. However, there are several multi-locus correlated regions in our genetic map, where markers on one chromosome are in linkage disequilibrium (LD) with markers on another chromosome (Supplementary Figure 7). Therefore, genes with cis-eQTL that are physically within these LD regions may have spurious trans-eQTL on other chromosomes. Since previous results suggested that cis-eQTL were pervasive and of large effect in HAL2-FIL2 crosses[44], we scanned for QTL peaks, conditioning on the additive covariates of treatment and the cis-eQTL genotype.

Both additive and QTL-by-treatment (GxE eQTL) scans were accomplished using the Haley-Knott algorithm on Hidden Markov Model multipoint genotype probabilities. The lowest genome-wide error rate-controlled empirical P-value (1000 permutations) for each transcript abundance phenotype was multiple-test corrected using the Q-value approach[48]. The maximum significant ($\alpha = 0.05$) empirical P-value was enforced as the trans-significance threshold. Only trans-eQTL LOD peaks that were more significant than this threshold were retained. Since cis-eQTL were not subject to genomic scans, we employed a standard Q-value transformed ANOVA $\chi^2$ P-value threshold, retaining cis effects with FDR-corrected P-values ≤0.05. Both cis- and trans-GxE effects were determined by comparing the additive and full models via likelihood ratio tests. The resultant P-values were FDR corrected in an identical manner as the additive cis-eQTL test. Finally, we dropped any cis- or trans-eQTL with maximum LOD scores <2.

**Genetic map construction**. To construct a linkage map, we calculated the ratio of HAL2/(HAL2 + FIL2) allele-specific expression (ASE) for 15,082 gene models with >5 ASE counts in each parent. For each chromosome and F2 genotype, we conducted a 20-gene overlapping sliding window calculation of the proportion of HAL2-ASE. We expected clear tri-modality of HAL2 allele frequency within each window, where the HAL2 homozygotes were ~100% HAL2 alleles, the heterozygotes were ~50%, and the FIL2 homozygotes were ~0%. Given the F2 population, the three peaks should be in approximately a 1:2:1 segregation ratio. Windows that did not conform to these expectations were dropped from the analysis.

To make calls, we first needed to develop thresholds that distinguished the three genotypes. Since the relative bias of allele frequency towards either parent in the heterozygote varied based on the markers in the window, a static threshold was not appropriate. Instead, we applied a dynamic threshold that separated the trimodal distribution of alleles into three bins. This dynamic thresholding eliminated library-specific biases towards either parent. The most common call (majority vote) in each 20-gene window was used as the genotype call. Calls were concatenated across libraries and chromosomes to produce a single genotype matrix.

Sliding window majority vote markers were processed to produce a saturated and non-redundant genetic map. Markers were first clustered into groups with identical genotypes. The marker with the least missing data was retained for each cluster of identical markers, resulting in a matrix of 3555 unique markers. We then clustered pairs of markers with ≤0.005 recombination fractions, retaining the marker that minimized missing data and segregation distortion. The physical assembly position was used to order markers and form linkage groups. After estimating a genetic map with the Kosambi mapping function[47], we culled markers within 0.5 cM, retaining the marker with lower segregation distortion. Finally, we tested for potentially problematic markers by assessing the genetic map size before and after dropping each marker. Those markers that resulted in an expansion of the map >1 cM were culled. The final map contained 1519 markers.

Given the approach of genotyping via allele-specific expression counts, we sought to test the genotyping error. To accomplish this, we assumed the marker order was correct, and iteratively checked parameters of map expansion across a range of assumed genotyping errors[49]. For all chromosomes, the −log likelihood of the genetic map was maximized at $1 \times 10^{-6}$ (Supplementary Figure 8), indicating that we would expect ≤1 genotyping error among all 200,430 marker-by-individual combinations in our genetic map.

**Tests for selection on eQTL**. To test for selection on the evolution of gene expression, we employed a sign test where the odds ratio of cis- and trans- up- and down-regulatory evolution was tested via a Fisher's exact test. Directional effects of each QTL were calculated using the fitqtl function in R/qtl. Estimates of QTL effects were calculated as the t-statistic of replacing the HAL2 allele with the FIL2 allele at a given locus. Deviations from the expected odds ratio (1:1) imply non-neutral evolution[29,50]. We also tested the odds ratio against the overall prevalence of antagonistic and reinforcing effects across the genome. This comparison allowed for inference of outlier regions relative to the genome average of cis-trans allelic effects.

**Promoter and coding DNA sequence variant annotation**. Reciprocal best hit (RBH) orthologs are defined as those pairs of genes where each is the other best matching sequence when compared to the alternate genome. Here we tested for 1:1 RBH orthologs by finding the best HAL2(query):FIL2(target) and FIL2(query):HAL2(target) protein BLAT[43] scores. The genes where both BLAT runs produced the same set of maximal gene pairings were the initial set of RBH orthologs. Given the known high degree of synteny, we then culled this set to include only pairings where both genes were on the same chromosomal region, allowing for 20% of the total chromosome length as buffer.

To annotate coding DNA sequences (CDS), FIL2 CDS sequences were aligned against corresponding HAL2 sequences using Minimap2[51]. The resulting alignment file was subset to 26786 HAL2-FIL2 orthologs. A pileup-formatted file was generated using SAMtools[52] mpileup utility. VarScan2[53] was used to call variants (SNPs and INDELs). Those variants were then annotated using SnpEFF[54]. SNP and INDEL variant annotations were described as synonymous, moderate effect (e.g. missense mutation), high effect (e.g. premature stop), or insertion/deletion. All genes were classified as either having or lacking significant CDS evolution. Significant genes had at least one moderate or larger effect SNP or INDEL, while non-significant genes were monomorphic, or had only low effect variants.

We annotated transcription factor binding affinity (TFBA) of the promoter sequences (defined as 2 kb upstream of the transcriptional start site) by extracting and comparing relative binding affinity of a set of 35 transcription factor families. Transcription factor binding sites (motifs) were downloaded from the JASPAR2016 plant database[55] as positional frequency matrices. Affinity of each motif was scored for each promoter sequence and differential binding affinity was tested using the R package PWMEnrich[56]. Binomial tests for equal affinity were performed for each gene and transcription factor. Significance was determined by controlling for global false discovery rate at $\alpha = 0.05$ by transforming the binomial P-values using the Benjami-Hochberg method. All genes were classified as either having or lacking significant TFBA evolution. Significant genes had at least one transcription factor with a significant binomial test while all transcription factors in non-significant TFBA genes had FDR-corrected binomial P-values ≥0.05.

In many cases, we sought to compare the relative prevalence of genes with and without TFBA and CDS evolution. To accomplish this, we employed a Fisher's exact test, reporting the odds ratio and exact P-value.

**Population genetic analyses**. We re-sequenced 94 P. hallii genotypes; 78 var. hallii, and 15 var. filipes. We selected more hallii samples than filipes as the geographic range of hallii is far larger than that of filipes. If identical sample sizes were used, we would be comparing geographically proximate filipes and geographically distant hallii. Since there is at least some isolation-by-distance in P. hallii[16], we instead opted to choose sample sizes that generally reflect the relative range size of each variety.

Illumina HiSeq2500 2 × 150 reads were mapped to the HAL2 genome reference via bwa-mem[57], followed by samtools[52] to sort the bam and create a multisample mileup[52]. Single nucleotide polymorphisms (SNP) and insertion-deletion (indel) polymorphisms were called via the Varscan 2.4.0[53] pipeline as above, with a minimum coverage of 8, and minimum reads of 4. A binomial test was used to confirm the allelic state (homozygous or heterozygous) at a P-value of 0.05. SNPs within 25 bp of a repeat were removed. A minimum presence of allele in 90% samples was required for further analysis.

Population structure was estimated using fastStructure[58] and TESS[59] with a subset of 50k markers, that were LD pruned (parameters: --indep-pairwise 50 50 0.5) in plink[60]. A single sample with a membership coefficient (qi) of <0.7 was considered admixed.

To infer demographic history, samples were phased (via SHAPEIT[17]). Sites were filtered for unique 24mers to minimize spurious variant calls. These data were input into MSMC2[17] to estimate effective population size and divergence using the following parameters: 4 haplotypes for each subpopulation, skipping ambiguous sites and a time segment pattern = '10*2 + 20*5 + 10*2' and an estimated rhoOverMu of 0.25. We estimated rhoOverMu as 0.25 as the mean value from 100 iterations without the fixed recombination parameter of five sets of four haplotypes in each subpopulation and averaged them. To estimate scaled times were converted to years assuming a generation time of 1 year and a mutation rate of 6.5*10−8. 50 bootstrap datasets obtained using MSMC tools, and simulations using the same parameters above were performed for each of the estimate to get the confidence intervals for population size estimates.

For the phylogenetic analysis, we first defined syntenic blocks to obtain an outgroup for the analysis. To define syntenic blocks, CDS DNA sequences were pulled and reciprocally mapped via blat[43] between the P. hallii genomes, Setaria viridis, Sorghum bicolor, and Switchgrass. Mappings with scores >100 and within 50% of the maximum mapping score for each gene were retained. Two-dimensional k-nearest neighbor clustering was employed to cull mappings to regions of dense hits in both genomes. For each chromosome pair, we chained mappings together using a traveling salesperson problem solver, which can be used to infer marker order in genetic mapping[61]. These chains were split if gaps >500 kb existed, and re-chained if two blocks were adjacent. We parsed 30 large syntenic blocks, which contained 326k variable sites across all non-admixed libraries. A random set of 50 K SNP were used for ML tree construction in RAXML[62].

**Dating divergence time**. OrthoFinder[41] was used with default parameters to identify clusters of single copy genes in 7 species along with *Arabidopsis* as outgroup. Of the 3311 ortholog sets, we randomly chose 100 sets and aligned them in Clustal-omega[63]. We used a HKY substitution model with 4 gamma rate categories and an uncorrelated log-normal relaxed clock model in BEAST2 (v2.4.7)[64] to estimate the divergence times. A calibration time of 52 Mya was used for *Brachypodium* and *Sorghum* split. We performed four independent runs, each with 10 million burn-ins and 50 million chain length, saving every 5000 chains. LogCombiner[64] was used to combine trees that have an effective sample size of at least 200. The consensus trees were further estimated in TreeAnnotator[64] using mean and median node heights.

**qPCR validation of RNA sequencing**. For quantitative RT-PCR, we isolated total RNA from the leaves using an RNA extraction kit (TRIzol reagent, Invitrogen). About 1 μg total RNA was reverse-transcribed using SSII reverse transcriptase (Invitrogen) in a volume of 80 μl to obtain cDNA. We used primers *PhABO3q*-F (5′-CCGCATACTGGATCTCACAA-3′) and *PhABO3q*-R (5′-TCTGAATC-CAGTGGCACATC-3′) for amplifying the transcript of *PhABO3* and *PhUBIq*-F (5′-TCATTTCGATGCTCTCTAGTCC-3′) and *PhUBIq*-R (5′-TTCCTGA-GATTTGCGAACAATG-3′) for ubiquitin-conjugating enzyme (*Pahal.3G250900*) as the internal control. We carried out quantitative RT-PCR in a total volume of 10 μl containing 2 μl of the reverse-transcribed product above, 0.25 μM gene-specific primers and 5 μl LightCycler 480 SYBR Green I Master (Roche) on a Roche LightCycler 480 II real-time PCR System according to the manufacturer's instructions. The measurements were obtained using the relative quantification method[65].

**Recombinant inbred line field experiment**. 294 $F_7$ individuals of the *HAL2xFIL2* recombinant inbred line (RIL) population[34] were grown at Brackenridge Field Laboratory (BFL, Austin, TX, USA) in spring and summer of 2016. Seeds were germinated in the BFL greenhouse on 29-Feburary and 3-March 2016. For each RIL line, seed coats for 40 seed were removed by scarification, and seeds were germinated in wet aquarium sand on petri plates. Sand was rewetted as necessary with Miracle Gro liquid fertilizer mix. Plate locations were randomly cycled daily in the greenhouse to standardize growth conditions. Germination occurred between 3–5 March and on 7–10 March. The healthiest seedlings were transplanted from sand to 32-cell trays to generate at least eight healthy replicates per line. Transplants were thinned to the most robust individual per cell between 23–29 March, and 294 lines with eight healthy individuals were selected for inclusion in the field design. On 7 April, plants were randomized within trays into the field experiment layout. To standardize growing conditions, trays were also cycled once in the greenhouse between transplanting and randomization.

RIL individuals were transplanted in the field on 14–15 April in 32 interspersed wet and dry treatment beds, three individuals wide and 26 individuals long. These blocks were surrounded by a border row of *FIL2* individuals. Individuals were spaced by 0.5 m along the length of the bed, and by 1.25 m along the width of the bed and between beds. Four individuals from each RIL line were randomized into both the wet treatment and the dry treatment beds to make a randomized split-plot design. Transplanting survival was very high, but nine plants were replaced in mid-May due to transplanting death. The entire field was irrigated on 14, 15, and 20 April to reduce transplantation shock. Thereafter, the wet treatment beds were frequently irrigated between 17-June and 8-August in order to achieve >50% soil volumetric water content. The dry treatment beds were irrigated at roughly 1/5th the frequency, at a level sufficient to achieve 30% soil volumetric water content, as measured with a Campbell Scientific CWS655 soil moisture probe. We measured two phenotypes: leaf relative water content (RWC = (weight$_{fresh}$−wt$_{dry}$)/(wt$_{turgid}$−wt$_{dry}$)) and leaf chlorophyll content (SPAD, using a SPAD 502 chlorophyll meter). Phenotypic tissue harvest and measurements took place from 11–15 July 2016 between 9AM and 1PM.

The best linear unbiased predictors were calculated for each genotype-by-treatment combination, where bed was treated as a random effect in the split plot design. The maximal position of each trans-eQTL hotspot was extracted and converted from the *HAL2* coordinate system of the $F_2$ genetic map to the *FIL2* (v2.0) coordinate system of the RIL genetic map[34]. The multipoint genotype probability for the proximate RIL marker to each hotspot was inferred via a hidden Markov model, as implemented in R/qtl's calc.genoprob function. Finally, we applied single marker tests for marker-phenotype associations between RWC and SPAD in the three conditions (wet, dry, plasticity [wet–dry]) as implemented in R/qtl's fitqtl function.

## Data availability

All RNA and resequencing reads have been deposited in the NCBI Short Read Archive (https://www.ncbi.nlm.nih.gov/sra). Bioprojects, sample IDs, and metadata can be found in Supplementary Data 1 and 7. Both genome assemblies and annotations are available through phytozome (https://phytozome.jgi.doe.gov). The assemblies have also been deposited on Genbank (https://www.ncbi.nlm.nih.gov/genbank) under BioProjects PRJNA251785 (*HAL2*) and PRJNA250527 (*FIL2*). All statistical, QTL mapping, and visualization functions were implemented in R 3.4.3 and have been compiled into an R package stored on github [https://github.com/

jtlovell/qtlTools]. Additional supporting data is provided as Supplementary Data. Details are provided in the Description of Additional Supplementary Files and the Reporting Summary. The source data underlying Figs. 1A-B, 2A-D, 3B-C and Supplementary Figure 3–7 are provided as a Source Data File. Source data for Fig. 3A, D and Supplementary Figure 1–2 are found in Supplementary Data 7 and Supplementary Data 2, respectively. A reporting summary for this Article is available as a Supplementary Information file.

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

## Acknowledgements

This project is the result of nearly a decade of field and laboratory work, which was accomplished with the help of J. Heiling, A. Asmus, C. Purmal, D. Des Marais, J. Reilley, T.S. Quedensley, B. Whitaker, K. Hernandez, E. Milano, M.J. Aspinwall, S.H. Taylor, and other members of the Juenger lab at UT Austin. We thank the UT Brackenridge Field Lab and the Ladybird Johnson Wildflower Center for providing support and field sites for experiments. Y. Stuart, S. Sen, K. Broman, D. Hoover, R. Williamson, R. Holmer, E. Schranz, and D. Bolnick provided advice on methods and presentation. Computational analyses were conducted, in part, on the Stampede system with allocations from the Texas Advance Computing Center. We thank Dan Sisco and Bill LaRochelle from 454/Roche for early data contributions to the *filipes* reference and Michael Kertesz for early access to Moleculo library technology. We thank collaborators at the Donald Danforth Center and the Joint Genome Institute for access to the prepublication genome and annotation of *Seteria viridis* to use in the outgroup analysis. This work was funded by NSF/IOS-1402393 (J.T.L.), NSF/IOS-1639872 (J.T.L./T.E.J.), NSF/IOS-0922457 (T.E.J.), NSF/IOS-1444533 (T.E.J.), DOE/DE-SC0008451 (T.E.J.), DOE/DE-SC0018409 (D.B.L.), DOE/DE-FC02-07ER64494 (D.B.L.), and USDA/NIFA 2011-67012-309969 (D.B.L.). E. V.S. was supported in part by the Russian Government Program of Competitive Growth of Kazan Federal University. The work conducted by the U.S. Department of Energy Joint Genome Institute is supported by the Office of Science of the U.S. Department of Energy under Contract No. DE-AC02-05CH11231.

## Author contributions

All authors contributed extensively to this project. Genome sequencing, assembly, and annotation were accomplished by K.B., C.D., A.L., C.P., S.S., Y.Y., M.Z., J.J., J.G., and J.S.; J.T.L., S.M., A.S., J.J., S.P.G., B.G., J.S., and T.E.J. conducted statistical analyses. DNA isolation and BAC library generation were accomplished by J.D.T. and K.D.; J.B., B.C., E. S., A.K., J.D.P.M., A.M., X.W., J.T.L., D.L., and T.E.J. designed and executed the experiments. J.T.L., J.S., and T.E.J. prepared the manuscript with input from all authors.

## Additional information

**Competing interests:** The authors declare no competing interests.

