## [Peer Review File · Nature Communications]

Reviewers' comments:

Reviewer #1 (Remarks to the Author):

Overall I found this to be an impressive study using innovative methods to address a key question. I have a few questions and suggestions.

1. It sounds like half of the F2s were profiled with RNA-seq 24 hrs after watering, and the other half without watering. This should result in three sets of gene expression "traits": with watering, without watering, and the difference between these (GxE). I could not find anywhere that said which one(s) of these were used for eQTL mapping. In Fig 3, which condition(s) are these eQTLs from? Are these only the GxE ones? And exactly how were the GxE expression traits defined— as the log ratio of normalized expr levels for each gene in the two conditions?

2. I see why the authors did allele-specific mapping for their linkage map, but for quantifying expression in the F2s, were allele-specific reads treated differently from those mapping to both parental genomes? If so, why? (Standard eQTL mapping just takes the total expression level of each gene in each F2 together with genotype; some new methods like RASQUAL do incorporate ASE, but this was not used here)

3. "Cis-regulated genes were far more likely to have accumulated mutations at both non-synonymous and transcription factor binding sites (both $P < 1 \times 10^{-16}$, Tables S12-13) than genes without cis-eQTL — evidence that cis-eQTL are driven by proximate regulatory loci, and not by genetically linked, but physically distant, regulatory loci." Here the authors seem to be assuming that the higher variation at cis-regulated genes is what is causing their cis-effects, but this need not be the case. Genes with cis-eQTLs are generally less constrained across all evolutionary timescales (which makes sense— any mutations impacting cis-regulation of the most important/conserved genes will be removed by purifying selection, so the observed cis-eQTLs will mostly target less constrained genes).

4. I didn't understand this sentence on line 273: "The threshold to call each genotype was determined as the least-commonly observed ratio of HAL2 ASE / total ASE above and below 50%." Perhaps giving an example would help.

5. Fig 3e: Testing for an imbalance in reinforcing/antagonistic effects within each trans-eQTL hotspot is a nice idea. From close inspection of the top panel, it looks like there are just a few more reinforcing than antagonistic in *3a (perhaps ~25 reinforcing vs ~20 antagonistic). This would only be significant as indicated at $p < 0.01$ if using an expectation of < 0.5 , so I'm guessing the authors used the genome-wide prevalence of reinforcing/antagonistic effects as their binomial expectation. However if this genome-wide prevalence is driven by stabilizing selection, as the authors propose, then a more appropriate null would be 0.5 for the binomial test; otherwise the test may just reveal less stabilizing selection on these genes (compared to the genome average), rather than more positive selection. It's a subtle but important difference for the interpretation of what's going on at the *3a locus.

6. It would be very interesting to see if there is any biological hypothesis that can be made about the reinforcing or antagonistic effects, using GO term enrichment. I would suggest testing the following gene sets: 1) all reinforcing; 2) all antagonistic; 3) FIL2-biased reinforcing; 4) HAL2-biased reinforcing. In all cases the set of all genes with both cis- and trans-eQTLs should be used as the background, to control for any functional enrichments within the eQTL targets as a whole. For example, this type of approach identified the ERG pathway in ref 27.

7. I especially like the physiological validation of the *3a hotspot. The effects however seem pretty weak, and wouldn't survive multiple test correction for six treatment-trait combinations. How does *3a compare to the rest of the genome? In a LOD score plot, would *3a stand out?

8. There is a methods section on inferring candidate genes driving trans hotspots, but the only mention of a candidate gene I can find is for ABO3. Was ABO3 expression correlated with expression of *3a target genes, as expected if a cis-eQTL for ABO3 is driving the hotspot? And if you have midday leaf water potential for the same F2s as the expression data, is ABO3 expression or the *3a genotype predictive of this trait?

Reviewer #2 (Remarks to the Author):

Lovell et al. produced chromosome scale assemblies of an upland (drought tolerant) and lowland switchgrass ecotype and map drought related eQTL using a large F2 mapping population constructed from these parents. This manuscript builds on previous work by this group mapping expression and regulatory dynamics in an F1 population of the same parents (HAL2 and FIL2; Lovell et al, Genome Research 2016). As expected, the majority of eQTL map to cis variants. Trans-eQTL are distributed non-randomly across the genome and a hotspot on Chromosome 3a is associated with variation in drought tolerance, and this interaction was validated using a RIL population. These switchgrass genomes will be of enormous use to the grass comparative genomics and biofuel research communities and these findings represent a major advance in our understanding of complex traits such as drought tolerance. I have a few comments I feel should be addressed to strengthen this manuscript prior to publication.

Major comments:

1. Line 90: HAL2 and FIL2 have near complete collinearity, and only 4% of the PAV genes are completely lost in one of the genomes. It is possible the other 96% of genes with detectable homology between ecotypes were mis-annotated because of residual indel errors in the final PacBio based genomes. It would be useful to validate that these are true pseudogenes by aligning RNAseq or Illumina WGS data to detect residual indels. A more simple approach would be to see if they have detectable expression.

2. Comparative genomics analyses between the two switch grass genomes are minimal outside of simple PAV detection. A pair of reference quality genomes have not been compared in detail for any species, and this is a unique opportunity. More details on fine scale changes in microsynteny, and repetitive element dynamics would be interesting.

3. It is unclear how the RIL population was genotyped and how the 3a trans-eQTL haplotype was identified in this population. Were QTL for RWC and chlorophyll content mapped? This is also unclear. The correlation between this trans hotspot and drought tolerance is not particularly strong. Chlorophyll content is usually correlated with drought tolerance, but the magnitude of change is not reported here. The distribution of RWC and chlorophyll content in this population should be provided.

4. Line 129: The finding that promoters are more conserved in genes with trans-eQTL than cis-eQTL is interesting, but this data is not presented anywhere. More details should be included.

Minor:

The title is not really reflective of what was done in this study, and on first glance, makes this seem like a simple genome release paper.

I would suggest adding some of the details from the supplementary note 1 to the methods of the paper. In particular, methods for pseudomolecule construction and polishing using Illumina data.

Line 253 Should be flash frozen

Reviewer #3 (Remarks to the Author):

Title: The *Panicum hallii* genome reveals molecular adaptation to drought

Authors: John T. Lovell^{1,2}, Jerry Jenkins¹, David B. Lowry³, Sujan Mamidi¹, Avinash Sreedasyam¹, Xiaoyu Weng², Kerrie Barry⁴, Jason Bonnette², Brandon Campitelli², Chris Daum⁴, Sean Gordon⁴, Billie A. Gould³, Anna Lipzen⁴, Alice MacQueen², Juan Diego Palacio-Mejia², Christopher Plott¹, Eugene V. Shakirov^{2,5}, Shengqiang Shu², Yuko Yoshinaga⁴, Matt Zane⁴, Daniel Rokhsar⁶, Jane Grimwood¹, Jeremy Schmutz^{1,4*}, Thomas E. Juenger^{2*}

This manuscript “*Panicum hallii* genome reveals molecular adaptation to drought” by Schmutz, Juenger, Lovell and colleagues is a monumental achievement that combines genome sequencing, field expression analyses, eQTL, and physiological ecology to elucidate regulatory expression evolution and candidate regulatory gene variants in drought adaptation in upland (dry *P. hallii* var *hallii*) and coastal/riparian (mesic *P. hallii* var *filipes*) varieties of the C4 model grass *Panicum hallii*. It should be of great interest to readers of *Nature Communications* as it comprehensively addresses molecular regulatory mechanisms (cis, trans eQTL) of drought response and could shed new light on potential genetic engineering of regulatory pathways that might increase drought tolerance in perennial grasses in increasingly water-stressed climates. The research is scientifically sound and results are adequately discussed within context of the literature. This is a very strong team with notable expertise in genome sequencing (Schmutz and Jenkins), eQTL (Juenger) and physiology (Lovell) and specifically with *P. hallii* ecology and adaptation (Lowry). The team’s combined talents have been brought to bear on the careful genomic dissection of regulatory mechanisms of drought response in these two contrasting grass varieties in a study that is almost unprecedented in its comprehensive scope. For these reasons, it meets the high criteria for publishing in *Nature Communications* provided the authors make some revisions.

In spite of my enthusiasm and appreciation for this research, several points need to be clarified and/or justified to make the manuscript even more compelling.

- 1) The abstract seems overreaching in terms of this study discovering drought tolerance mechanisms that can be used to solve the US bioenergy problems (lines 24-30). This seems like a bit of an oversell and speculative, at best. The research makes a substantial contribution in understanding molecular adaptation of drought tolerance in these two contrasting grass varieties and does not need to overreach to solve bioenergy problems to make it compelling. The suggestion that the research provides the basis for genetic engineering of regulatory networks also seems a bit of an overreach. The research makes a significant contribution in its own right, without having to rely on its importance for biofuel production. Furthermore, applying this knowledge from the small diploid genome of *P. hallii* to C4 biofuel grasses with large polyploid genomes such as switchgrass *Panicum virgatum* is a daunting task and will take many years.

- 2) All of this research is based on the sequencing, eQTL and physiology studies of two individuals (HAL2 and FIL2). How does the initial small sample size of *var filipes* affect the downstream results and interpretation? For example, the authors initially deep sequence 94 plants total, only one plant per population (79 *var hallii* and 15 *var filipes*, line 69). In figure 1A, the genotyping assignments show that in fact only 10 are classified as *var filipes* (blue circles) while the remaining 84 (yellow, orange or brown circles) are *var hallii* dispersed from across TX, NM and AZ. Is the deep sequencing of a relatively small number of *var filipes* plants (seemingly 10) adequate to provide enough representation of the range of variation in *var filipes* populations? For example, if the authors had deep sequenced 40 *var filipes* populations and selected the most divergent one from the basal *P hallii var hallii*, how might this have impacted results? Please provide justification that the 10 *var filipes* individuals adequately represent the standing variation in the populations, or justify why the small sample size was chosen and its impact on the interpretation of the results.
- 3) Are there genes in coding regions that might also shed light on differences between *var hallii* and *var filipes*? Of the 4% of Presence Absence Variants (line 90, about 120 genes), what are these genes and can they help explain the differences between *var hallii* and *var filipes*?
- 4) The authors do a commendable job at describing the physiological differences in the two varieties in response to drought and connecting that to eQTL and trans hot spots. Are there are phenotypes besides physiological parameters that could also shed light on differences between varieties? For example, timing of flowering or reproductive effort? Early flowering is known to be a response to drought. Similarly, are there other phenotypes related to morphology that could be included such as height, number of tillers, blade width, leaf area or biomass? This is a clear gap in the study. Furthermore, if the authors want to relate to potential bioenergy production, results need to be related to plant biomass (necessary for bioenergy) not just physiology. This is another reason to temper the implications of this study for bioenergy as currently emphasized in the abstract.
- 5) The authors first emphasize the importance of cis-regulatory QTL on lines 121-133 but then in lines 134-139 highlight that the globally rare trans-eQTL are the real key to GXE effects and adaptive trans-regulatory evolution between varieties. They then proceed to convincingly to demonstrate the three trans-hotspots. The earlier discussion about the cis-regulatory QTL seems like a distraction from the more compelling trans-eQTL hot spots. Can the authors temper the cis-eQTL part and keep the reader focused on the trans eQTL hot spots?
- 6) Lines 175-180. Please provide a better description and definition of antagonistic vs reinforcing effects.

- 7) Revise Fig. 1B so the reader can actually see the pie charts showing genotyping assignment.
- 8) Finally, I was frustrated by the format of the “supplemental tables” provided. The manuscript would benefit greatly from actual summary tables, not just data sets. Maybe this was a problem with uploading, but summary tables for key data are clearly needed, not just the data sets. Essentially, the reader is expected to take the authors on face value that their summary results are correct or are required to do the analyses for themselves and make a summary table. This really needs to be addressed. It seems likely the authors must have these summary tables prepared and should include them. Furthermore, on the data set files, there is no description of columns. Also there is no code provided in supplemental for the analyses. Furthermore, on line 229, PERTRAN was used based on Shu unpublished. How would the reader access this?

Reviewer #4 (Remarks to the Author):

This is an interesting and well-done analysis of a diverse and well-designed set of experiments. I found the work to be compelling and much of the interpretation to be insightful and thought-provoking. The datasets created for this are substantial and quite complementary, and represent an important resource for future study, so it is great to see that the authors have already made their data public. This is truly an impressive set of work, and the researchers should be commended for their study design, analysis, data archiving, figures, and writing.

My only quibble is with one aspect of the interpretation. One alternate way of thinking about the results is that the GxE may not really be due to regulatory genes, at least as we typically think of them, regulating the genes in question, e.g. through series of kinases and transcription factors, etc. In fact, what might be going on is that some of the major QTLs identified are instead reducing the effects of the stress through various mechanisms, which would of course affect expression of a broad range of genes but not through typical regulatory mechanisms, but instead by changing the entire cellular environment (by ameliorating the stress itself). This kind of alternate explanation is often ignored in RNAseq experiments, but could nonetheless be important in this study.

My guess is that the genes identified in this study are a mix of some genes that really are directly or indirectly regulated by ABO3 (or something similar, but that is a convincing candidate gene) and some genes that are not really regulated but are simply changing expression due to the changing cellular environment. The data presented suggests that at least many of the effects identified are more in line with a true regulatory effect of something like ABO, though, so much of the interpretation written is likely to be correct. I think the author's claim that they have identified a 'global regulatory QTL' is actually right, and it's exciting that they even have a very strong candidate gene in one of their QTL regions. However, it would be important to clarify that not all of the genes identified as changing in expression are necessarily regulated, in the normal way, by this QTL - there are likely to be many genes that are not part of any true regulatory pathway, but are instead changing their expression due to the reduced stress or other differences in the cellular environment.

Essentially, what I am advocating is to think about what it means to be a 'regulatory QTL' and to define that more carefully before going into the rest of the analysis. Other than this clarification, the paper is excellent and should be published with only minor revisions.

Reviewer #1

It sounds like half of the F2s were profiled with RNA-seq 24 hrs after watering, and the other half without watering. This should result in three sets of gene expression “traits”: with watering, without watering, and the difference between these (GxE). I could not find anywhere that said which one(s) of these were used for eQTL mapping. In Fig 3, which condition(s) are these eQTLs from? Are these only the GxE ones? And exactly how were the GxE expression traits defined— as the log ratio of normalized expr levels for each gene in the two conditions?

- We were not clear in the description of the eQTL methods and have improved and extended the description in both the results [175-179, 188-192] and methods [419-437] sections, which we hope will ameliorate this confusion.
- In short, since each genotype (F2 line) only received a single treatment, we could not take the difference between treatments (as would be done in a replicated RIL eQTL, e.g. Lowry et al. 2013). Instead, GxE must be assessed by a linear model where QTL genotype (AA/Aa/aa) and treatment are crossed and the significance of the interaction term is assessed by a likelihood ratio test.

I see why the authors did allele-specific mapping for their linkage map, but for quantifying expression in the F2s, were allele-specific reads treated differently from those mapping to both parental genomes? If so, why? (Standard eQTL mapping just takes the total expression level of each gene in each F2 together with genotype; some new methods like RASQUAL do incorporate ASE, but this was not used here)

- Again, the methodological description of the linkage map was lacking. Like you thought, we did not do eQTL mapping using ASE. ASE was only used to build the genetic map. We now present the two different datasets as two completely separate method sections and present much more detail about genetic map building under its own subheading [lines 440-470]. This way ASE and total counts are not discussed in the same section, which we think will avoid confusion.
- Finally, we drop all discussion of ASE in the eQTL results. This was confusing, since we never actually perform eQTL analysis with these data except to discuss expression of the PAV genes.

“Cis-regulated genes were far more likely to have accumulated mutations at both non-synonymous and transcription factor binding sites (both $P < 1 \times 10^{-16}$, Tables S12-13) than genes without cis-eQTL — evidence that cis-eQTL are driven by proximate regulatory loci, and not by genetically linked, but physically distant, regulatory loci.” Here the authors seem to be assuming that the higher variation at cis-regulated genes is what is causing their cis-effects, but this need not be the case. Genes with cis-eQTLs are generally less constrained across all evolutionary timescales (which makes sense— any mutations impacting cis-regulation of the most important/conserved genes will be removed by purifying selection, so the observed cis-eQTLs will mostly target less constrained genes).

- This point is well taken and was not considered in the previous version. Now, we have worked to simplify the statements in this section and expand the interpretation of cis-eQTL regulatory sequences to include discussion of generally less evolutionary constraint [e.g. lines 219-223].

I didn't understand this sentence on line 273: "The threshold to call each genotype was determined as the least-commonly observed ratio of HAL2 ASE / total ASE above and below 50%." Perhaps giving an example would help.

- Yes, this is really confusing. In short, since we use allele frequency as our genotype, we need to threshold to produce 'hard' genotype calls. We did this dynamically, based on the distribution of allele frequencies. The previous method description was clearly not satisfactory. To resolve this issue, we have expanded the description to an entire paragraph [lines 448-454].

*Fig 3e: Testing for an imbalance in reinforcing/antagonistic effects within each trans-eQTL hotspot is a nice idea. From close inspection of the top panel, it looks like there are just a few more reinforcing than antagonistic in *3a (perhaps ~25 reinforcing vs ~20 antagonistic). This would only be significant as indicated at $p < 0.01$ if using an expectation of < 0.5 , so I'm guessing the authors used the genome-wide prevalence of reinforcing/antagonistic effects as their binomial expectation. However if this genome-wide prevalence is driven by stabilizing selection, as the authors propose, then a more appropriate null would be 0.5 for the binomial test; otherwise the test may just reveal less stabilizing selection on these genes (compared to the genome average), rather than more positive selection. It's a subtle but important difference for the interpretation of what's going on at the *3a locus.*

- This is a crucial comment. Your interpretation is spot on. The previous analysis used the background cis-trans bias as the NULL proportion. Initially, we opted to assume a genome-wide neutral evolution NULL hypothesis, in which case the NULL distribution is the background cis-trans bias. However, after considering your comments here and discussing with colleagues, we agree with your interpretation: the NULL should really be 1:1 (probability = 0.5 in the binomial test). We have altered all results and interpretation [lines 287-297] and methods [lines 475-482] accordingly.
- In one case [line 293], we also present the comparisons to the background proportion, as we believe this will be interesting to readers.

It would be very interesting to see if there is any biological hypothesis that can be made about the reinforcing or antagonistic effects, using GO term enrichment. I would suggest testing the following gene sets: 1) all reinforcing; 2) all antagonistic; 3) FIL2-biased reinforcing; 4) HAL2-biased reinforcing. In all cases the set of all genes with both cis- and trans-eQTLs should be used as the background, to control for any functional enrichments within the eQTL targets as a whole. For example, this type of approach identified the ERG pathway in ref 27.

- We initially explored this route; however, we really don't have enough power to divide the affected genes into groups. For example, the 4 proposed groups would all contain < 20 genes (since not all genes have both trans and cis-eQTL). While we still considered presenting these results, in the end, we decided that 20 genes really is not enough to make statistically rigorous tests of enrichment.

*I especially like the physiological validation of the *3a hotspot. The effects however seem pretty weak, and wouldn't survive multiple test correction for six treatment-trait combinations. How does *3a compare to the rest of the genome? In a LOD score plot, would *3a stand out?*

- We are glad you brought this up. Our previous methods descriptions were confusing. Although we used a RIL mapping population, we did not actually do any genome-wide scans. For this particular analysis, we are not concerned about the genetic architecture of these traits. Instead, we are only interested in a specific test: does allelic variation at a trans-eQTL hotspots affect RWC and SPAD. We now provide much more detail about this method [lines 603-610]. To avoid confusion, we only reference the genetic map in the associated publication and do not mention QTL mapping at all when referencing the RIL population.

*There is a methods section on inferring candidate genes driving trans hotspots, but the only mention of a candidate gene I can find is for ABO3. Was ABO3 expression correlated with expression of *3a target genes, as expected if a cis-eQTL for ABO3 is driving the hotspot? And if you have midday leaf water potential for the same F2s as the expression data, is ABO3 expression or the *3a genotype predictive of this trait?*

- For clarity, we have dropped this analysis and methods section.

Reviewer #2

Line 90: HAL2 and FIL2 have near complete collinearity, and only 4% of the PAV genes are completely lost in one of the genomes. It is possible the other 96% of genes with detectable homology between ecotypes were mis-annotated because of residual indel errors in the final PacBio based genomes. It would be useful to validate that these are true pseudogenes by aligning RNAseq or Illumina WGS data to detect residual indels. A more simple approach would be to see if they have detectable expression.

Comparative genomics analyses between the two switch grass genomes are minimal outside of simple PAV detection. A pair of reference quality genomes have not been compared in detail for any species, and this is a unique opportunity. More details on fine scale changes in microsynteny, and repetitive element dynamics would be interesting.

- This is an important comment. Previously (when submitting to *Nature*), we were very constrained by space. Now with more room for text, we have dramatically expanded our presentation and analysis of comparative genomics between HAL2 and FIL2 [lines 114-172].
- While we do not think there are many indel errors in the PacBio assembly, there are certainly many non-biological reasons for the presence of PAV genes. As pointed out, one of these may be low expression support. We now present detailed sequence similarity analyses between annotated and un-annotated (PAV) genes [table 1] and the associated score of those genes. As you thought, many of the previously described PAV loci are simply due to thresholding ... many genes have very similar homologous sequence that are simply un-annotated in the alternate genome. We now treat these cases in a far more nuanced manner, presenting both gene model confidence and sequence similarity.

It is unclear how the RIL population was genotyped and how the 3a trans-eQTL haplotype was identified in this population. Were QTL for RWC and chlorophyll content mapped? This is also unclear. The correlation between this trans hotspot and drought tolerance is not particularly strong. Chlorophyll content is usually correlated with drought tolerance, but the magnitude of change is not reported here. The distribution of RWC and chlorophyll content in this population should be provided.

- Reviewer 1 also brought up this comment. We have re-written the entire methods and results for this section, highlighting that we did not actually perform genome-wide QTL analyses, but instead, simply leverage a mapping population to perform three separate single-marker tests of association. See above for more details.

Line 129: The finding that promoters are more conserved in genes with trans-eQTL than cis-eQTL is interesting, but this data is not presented anywhere. More details should be included.

- We have expanded this results section and now include a table dedicated to the presentation of these results (Table 2).

The title is not really reflective of what was done in this study, and on first glance, makes this seem like a simple genome release paper.

- We agree and have altered the title accordingly.

I would suggest adding some of the details from the supplementary note 1 to the methods of the paper. In particular, methods for pseudomolecule construction and polishing using Illumina data.

- We have significantly expanded the genome assembly methods section. We have also re-built the supplementary material. Supplementary note 1 is now far more readable and detailed.

Line 253 Should be flash frozen

- Fixed

Reviewer #3

*The abstract seems overreaching in terms of this study discovering drought tolerance mechanisms that can be used to solve the US bioenergy problems (lines 24-30). This seems like a bit of an oversell and speculative, at best. The research makes a substantial contribution in understanding molecular adaptation of drought tolerance in these two contrasting grass varieties and does not need to overreach to solve bioenergy problems to make it compelling. The suggestion that the research provides the basis for genetic engineering of regulatory networks also seems a bit of an overreach. The research makes a significant contribution in its own right, without having to rely on its importance for biofuel production. Furthermore, applying this knowledge from the small diploid genome of *P. hallii* to C4 biofuel grasses with large polyploid genomes such as switchgrass *Panicum virgatum* is a daunting task and will take many years.*

- This point is well taken. We have completely re-structured the abstract according to these comments. In particular, we have toned down the language regarding biofuels and *P. hallii*'s potential role in improving yield. We now focus more on understand drought-responses and gene expression regulatory networks.

*All of this research is based on the sequencing, eQTL and physiology studies of two individuals (HAL2 and FIL2). How does the initial small sample size of var filipes affect the downstream results and interpretation? For example, the authors initially deep sequence 94 plants total, only one plant per population (79 var hallii and 15 var filipes, line 69). In figure 1A, the genotyping assignments show that in fact only 10 are classified as var filipes (blue circles) while the remaining 84 (yellow, orange or brown circles) are var hallii dispersed from across TX, NM and AZ. Is the deep sequencing of a relatively small number of var filipes plants (seemingly 10) adequate to provide enough representation of the range of variation in var filipes populations? For example, if the authors had deep sequenced 40 var filipes populations and selected the most divergent one from the basal *P. hallii* var hallii, how might this have impacted results? Please provide justification that the 10 var filipes individuals adequately represent the standing*

variation in the populations, or justify why the small sample size was chosen and its impact on the interpretation of the results.

- We now detail the justification in the text [lines 515-519]. In short, var. filipes is narrowly geographically distributed. As such, we maintained similar spatial sampling density between the two varieties to not bias population structure assays. It is possible that very different sampling density would lead to different results in terms of the most divergent line, but in the end, what we really want is a representative genotype for each group.

Are there genes in coding regions that might also shed light on differences between var hallii and var filipes? Of the 4% of Presence Absence Variants (line 90, about 120 genes), what are these genes and can they help explain the differences between var hallii and var filipes?

- Really good question. How does PAV between FIL2/HAL2 extend to population structure between varieties? We have added some text describing possible results, but we do not undertake this analysis. We have future efforts planned to do more deep/longer read sequencing of the population samples. This data would be appropriate to ask about PAV, but the short reads we have now won't do it justice.
- We have however expanded our tests of PAV genes between HAL2 and FIL2. See response to reviewer 2's comment above.

The authors do a commendable job at describing the physiological differences in the two varieties in response to drought and connecting that to eQTL and trans hot spots. Are there are phenotypes besides physiological parameters that could also shed light on differences between varieties? For example, timing of flowering or reproductive effort? Early flowering is known to be a response to drought. Similarly, are there other phenotypes related to morphology that could be included such as height, number of tillers, blade width, leaf area or biomass? This is a clear gap in the study. Furthermore, if the authors want to relate to potential bioenergy production, results need to be related to plant biomass (necessary for bioenergy) not just physiology. This is another reason to temper the implications of this study for bioenergy as currently emphasized in the abstract.

- Definitely agree with all these points. However, in the end, the goal of this study is to understand the genetic basis of drought responses and not biomass. We have altered the abstract accordingly and discuss the lack of biomass-related phenotypes when interpreting the results and conclusions.
- As for phenology, we agree – timing of flowering can be a critical drought response. We therefore sought to control for this by conducting the experiment once all genotypes had reached anthesis. We now state this explicitly in the methods [line 398].

The authors first emphasize the importance of cis-regulatory QTL on lines 121-133 but then in lines 134-139 highlight that the globally rare trans-eQTL are the real key to GXE effects and adaptive trans-regulatory evolution between varieties. They then proceed to convincingly to demonstrate the three trans-hotspots. The earlier discussion about the cis-regulatory QTL seems like a distraction from the more compelling trans-eQTL hot spots. Can the authors temper the cis-eQTL part and keep the reader focused on the trans eQTL hot spots?

- We now add in discussion of the interplay between cis- and trans-eQTL expectations of each. In the end, we hope to convince readers that both are important by connecting the effects of the trans-eQTL back to cis-elements that likely underlie these effects.

Lines 175-180. Please provide a better description and definition of antagonistic vs reinforcing effects.

- We have expanded our discussion of these two effects [lines 278-287]

Revise Fig. 1B so the reader can actually see the pie charts showing genotyping assignment.

- We've made the pies a bit larger, hopefully this helps.

Finally, I was frustrated by the format of the "supplemental tables" provided. The manuscript would benefit greatly from actual summary tables, not just data sets. Maybe this was a problem with uploading, but summary tables for key data are clearly needed, not just the data sets. Essentially, the reader is expected to take the authors on face value that their summary results are correct or are required to do the analyses for themselves and make a summary table. This really needs to be addressed. It seems likely the authors must have these summary tables prepared and should include them. Furthermore, on the data set files, there is no description of columns. Also there is no code provided in supplemental for the analyses.

- This is a critical comment. We agree fully. To this end, we have added three in-text summary tables and seven supplementary summary tables. Additionally, we have expanded the captions for the large supplementary tables and included a header in each with column name descriptions.

Furthermore, on line 229, PERTRAN was used based on Shu unpublished. How would the reader access this?

- We have added Supplementary Note 2, which describes the PERTRAN pipeline.

Reviewer #4 (Remarks to the Author):

My only quibble is with one aspect of the interpretation. One alternate way of thinking about the results is that the GxE may not really be due to regulatory genes, at least as we typically think of them, regulating the genes in question, e.g. through series of kinases and transcription factors, etc. In fact, what might be going on is that some of the major QTLs identified are instead reducing the effects of the stress through various mechanisms, which would of course affect expression of a broad range of genes but not through typical regulatory mechanisms, but instead by changing the entire cellular environment (by ameliorating the stress itself). This kind of alternate explanation is often ignored in RNAseq experiments, but could nonetheless be important in this study. My guess is that the genes identified in this study are a mix of some genes that really are directly or indirectly regulated by ABO3 (or something similar, but that is a convincing candidate gene) and some genes that are not really regulated but are simply changing expression due to the changing cellular environment. The data presented suggests that at least many of the effects identified are more in line with a true regulatory effect of something like ABO, though, so much of the interpretation written is likely to be correct. I think the author's claim that they have identified a 'global regulatory QTL' is actually right, and it's exciting that they even have a very strong candidate gene in one of their QTL regions. However, it would be important to clarify that not all of the genes identified as changing in expression are necessarily regulated, in the normal way, by this QTL - there are likely to be many genes that are not part of any true regulatory pathway, but are instead changing their expression due to the reduced stress or other differences in the cellular environment. Essentially, what I am advocating is to think about what it means to be a 'regulatory QTL' and to define that more carefully before going into the rest of the analysis. Other than this clarification, the paper is excellent and should be published with only minor revisions.

- This is a very important point and one that we glossed over. To support our claim that it is a regulatory QTL, and not simply a physiological QTL that indirectly affects gene expression, we add a completely new QTL analysis, which demonstrates that there is no effect of leaf water potential at this QTL. This indicates that water status in plants with different alleles is largely similar (lines 247-257). While certainly other methods of physiological buffering besides plant water status are possible, these results provide more support for our interpretation.

REVIEWERS' COMMENTS:

Reviewer #1 (Remarks to the Author):

The authors have satisfied my concerns.

Reviewer #2 (Remarks to the Author):

The authors have addressed my previous concerns.

Reviewer #3 (Remarks to the Author):

This manuscript "The genomic landscape of molecular responses to natural drought stress in *Panicum hallii*" by Schmutz, Juenger, Lovell and colleagues is a monumental achievement that combines genome sequencing, field expression analyses, eQTL, and physiological ecology to elucidate regulatory expression evolution and candidate regulatory gene variants in drought adaptation in upland (dry *P. hallii* var *hallii*) and coastal/riparian (mesic *P. hallii* var *filipes*) varieties of the C4 model grass *Panicum hallii*. It should be of great interest to readers of *Nature Communications* as it comprehensively addresses molecular regulatory mechanisms (cis, trans eQTL) of drought response and could shed new light on potential genetic engineering of regulatory pathways that might increase drought tolerance in perennial grasses in increasingly water-stressed climates. The research is scientifically sound and results are adequately discussed within context of the literature. This is a very strong team with notable expertise in genome sequencing (Schmutz and Jenkins), eQTL (Juenger) and physiology (Lovell) and specifically with *P. hallii* ecology and adaptation (Lowry). The team's combined talents have been brought to bear on the careful genomic dissection of regulatory mechanisms of drought response in these two contrasting grass varieties in a study that is almost unprecedented in its comprehensive scope. For these reasons, it meets the high criteria for publishing in *Nature Communications*.

The authors have carefully made my suggested revisions and clarifications or have provided justifications for not making the revisions. Additionally, they have also adequately addressed the concerns of the other reviewers. This research makes a compelling contribution to our understanding of the genomic underpinnings of physiological drought response in *P. hallii* and will have applications to our understanding of drought response in other grasses. In my opinion, no further revision is necessary.

Reviewer #4 (Remarks to the Author):

These revisions are thorough and quite well done, and the manuscript is substantially improved, even though it started out quite good. I have no further suggested revisions.

REVIEWERS' COMMENTS:

Reviewer #1 (Remarks to the Author): The authors have satisfied my concerns.

*** *No recommended revisions****

Reviewer #2 (Remarks to the Author): The authors have addressed my previous concerns.

*** *No recommended revisions****

Reviewer #3 (Remarks to the Author): This manuscript “The genomic landscape of molecular responses to natural drought stress in *Panicum hallii*” by Schmutz, Juenger, Lovell and colleagues is a monumental achievement that combines genome sequencing, field expression analyses, eQTL, and physiological ecology to elucidate regulatory expression evolution and candidate regulatory gene variants in drought adaptation in upland (dry *P. hallii* var *hallii*) and coastal/riparian (mesic *P. hallii* var *filipes*) varieties of the C4 model grass *Panicum hallii*. It should be of great interest to readers of Nature Communications as it comprehensively addresses molecular regulatory mechanisms (cis, trans eQTL) of drought response and could shed new light on potential genetic engineering of regulatory pathways that might increase drought tolerance in perennial grasses in increasingly water-stressed climates. The research is scientifically sound and results are adequately discussed within context of the literature. This is a very strong team with notable expertise in genome sequencing (Schmutz and Jenkins), eQTL (Juenger) and physiology (Lovell) and specifically with *P. hallii* ecology and adaptation (Lowry). The team’s combined talents have been brought to bear on the careful genomic dissection of regulatory mechanisms of drought response in these two contrasting grass varieties in a study that is almost unprecedented in its comprehensive scope. For these reasons, it meets the high criteria for publishing in Nature Communications. The authors have carefully made my suggested revisions and clarifications or have provided justifications for not making the revisions. Additionally, they have also adequately addressed the concerns of the other reviewers. This research makes a compelling contribution to our understanding of the genomic underpinnings of physiological drought response in *P. hallii* and will have applications to our understanding of drought response in other grasses. In my opinion, no further revision is necessary.

*** *No recommended revisions****

Reviewer #4 (Remarks to the Author):

These revisions are thorough and quite well done, and the manuscript is substantially improved, even though it started out quite good. I have no further suggested revisions.

*** *No recommended revisions****